# Evaluating the Facial Esthetic Outcomes of Digital Smile Designs Generated by Artificial Intelligence and Dental Professionals

Gülsüm Ceylan [1,*], Gülsüm Sayın Özel [1], Gözde Memişoglu [1], Faruk Emir [2] and Sevgin Şen [3]

1   Department of Prosthodontics, School of Dentistry, Istanbul Medipol University, Istanbul 34083, Türkiye
2   Department of Prosthodontics, Gülhane Faculty of Dentistry, Health Sciences University, Ankara 06010, Türkiye
3   Independent Researcher, Bursa 16200, Türkiye
*   Correspondence: gulsumcyln@gmail.com

**Abstract:** This study evaluates the preference rates for smile designs created by professionals or by Artificial Intelligence (AI) among dentists, dentistry students, and laypeople. Four cases with symmetrical and asymmetrical features were selected based on the Facial Flow (FF) concept from the database of the Smile Designer app regarding anatomical facial points. Two smile designs were created for each selected case: one using Artificial Intelligence (AI) and one created manually. An online survey assessed participants' preferences for the different smile designs. The chi-square test "Pearson's and Fisher's exact test (P)" was used to analyze the survey data. A total of 628 people completed the study. Dentists preferred the manually-created smile design for the first three cases. For Case 4, dentists who used the Smile Designer program preferred the manually-created design (55.88%), while those who did not use the program preferred the AI-generated design (55.84%). There was a significant difference in esthetic perception between dentists and dental students ($p = 0.001$) and between dentists and laypeople ($p = 0.001$) for Case 1, only between dentists and dental students ($p = 0.003$) for Case 2, and only between dentists and laypeople ($p = 0.001$) for Case 3. Furthermore, we found that females ($p = 0.007$) and orthodontists ($p = 0.025$) had a higher preference for the AI-generated design in this case compared to males and other dental specialties for Case 3. While age, education level, and clinical experience did not significantly impact dentists' preference for manually-created or AI-generated smile designs ($p > 0.05$), our results suggest that there were some differences in preference for Case 3. Overall, our findings suggest that the use of AI-generated smile designs for symmetric faces is acceptable to both dentists and laypeople and can offer time-saving benefits for clinicians.

**Keywords:** Artificial Intelligence (AI); dental esthetic; digital smile design (DSD); esthetic perception





## 1. Introduction

One of the primary goals of dental treatment is to restore teeth in a way that meets the patient's needs and creates a natural-looking, esthetically pleasing smile [1]. Owing to technological advancements, esthetic dental materials and treatments are becoming more widely available [2]. The use of three-dimensional (3D) design technology to create personalized, natural-looking, and esthetically pleasing smiles is becoming increasingly popular in dental practices, and "Digital smile design" (DSD) programs are a key tool in this process. These programs allow the dentist and patient to plan treatment and provide a high level of convenience [3,4]. There are many DSD programs available on the market, and a common feature of all of them is that they evaluate the smile in the context of the entire face [5]. This requires a photograph in which the patient is naturally smiling [6–8].

The use of computer technology for the esthetic prosthetic treatment of patients is becoming increasingly common. Studies have shown that digitally-made designs are

effective in meeting patients' esthetic expectations and achieving a predictable, satisfactory treatment outcome [5,9]. One of the first studies to introduce digital smile analysis and design was conducted in 2002 [10]. Initially, programs such as PowerPoint, Keynote, and Photoshop were used for planning smile designs [5]. In the past, clinicians and technicians would perform esthetic analysis on printed photographs or plaster models obtained prior to treatment. These programs allow for the creation of esthetic reference lines on faces and smile photographs on the computer [4].

In 2008, the first DSD protocol was developed, which associated a series of facial, intraoral, and extraoral photographs with the face [11]. This allowed for a fully digital format transition, enabling the matching of 2D photographs with 3D digital models and the verification and improvement of 3D esthetic parameters [3,11–13]. Computer programs and software tools help with the planning, execution, and visualization of restorative, prosthetic, orthodontic, surgical, and multidisciplinary treatments, while enabling the digital planning and visualization of expected esthetic results before treatment [14]. With the increased use of DSD programs in clinical settings, many innovations have been made to make these programs more accessible and practical. One such innovation is the integration of artificial intelligence into the software [15].

The term "AI" originated in the 1950s and referred to the idea that machines can perform tasks that humans typically perform [16]. In the future, AI is expected to automate esthetic evaluation, smile design, and treatment planning processes [2,17,18]. One of the main claims of companies is that smile designs created by AI can be produced in a short period of time (seconds), compared to those traditionally created by physicians [15,18].

A key point in designing natural and harmonious smiles is that faces and smiles are not always symmetrical [19]. Various studies have shown a positive relationship between facial symmetry and beauty [20,21]. Facial Flow (FF) is defined as "the direction in which facial structures travel." The Facial Flow Line (FFL) can usually be towards the left or right side of the patient's face. FF is also defined as neutral in cases where the nose and chin tip go in opposite directions of the face, or in cases where the face is symmetrical [19]. Anatomical landmarks on symmetrical faces can be manually determined and used as reference points [20,22–24]. However, more research needs to be conducted on whether these landmarks, which are crucial in creating smile designs, can be detected on asymmetrical faces using AI. It is noted that the application of AI in healthcare is still immature and requires significant improvement [25]. AI works according to a specific algorithm, and misinterpretations can occur due to the limitations of those algorithms [26].

The aim of this study was to evaluate the esthetic perception of facially driven smile designs created manually or using AI. The initial hypothesis was that, in smile design programs, the method of selecting landmarks (manually or using AI) does not affect the esthetic perception of the resulting smile design, regardless of the symmetry of the face.

## 2. Materials and Methods

### 2.1. Model Selection

Digital Smile Design Program (Smile Designer App advanced v.1.0, Neuralp Yazılım Bilişim Ltd. Şti., Bursa, Turkey) was used to design smiles. To evaluate the efficiency of the AI mode, the 4 cases were selected based on the Facial Flow (FF) concept from the sample cases in the Smile Designer App's database. The selection process of the cases was done through a group discussion involving a team of dental professionals with experience in cosmetic dentistry. The cases were chosen based on their clinical relevance and representativeness of the different smile designs examined in the study.

The classification of cases was based on the relationships between reference points on the forehead, nose, and chin tip, which are significant in smile designs. Specifically, the trichion, glabella, subnazale, and menton points were utilized as references to determine the facial flow. By examining the axis formed by connecting these points, we identified four distinct cases: Case 1 represented a facial flow towards the right side, Case 2 represented a facial flow towards the left side, Case 3 represented a scenario where the nose and chin

pointed in different directions, and Case 4 represented a symmetrical face (Figure 1). To aid visual clarity, we utilized a color scheme in Figure 1, where the side with the facial flow direction was depicted in green, while the opposite side was shown in red. However, it is important to note that in neutral cases where the nose and chin tips were directed to different sides while maintaining facial symmetry, no green or red sides were indicated [19].

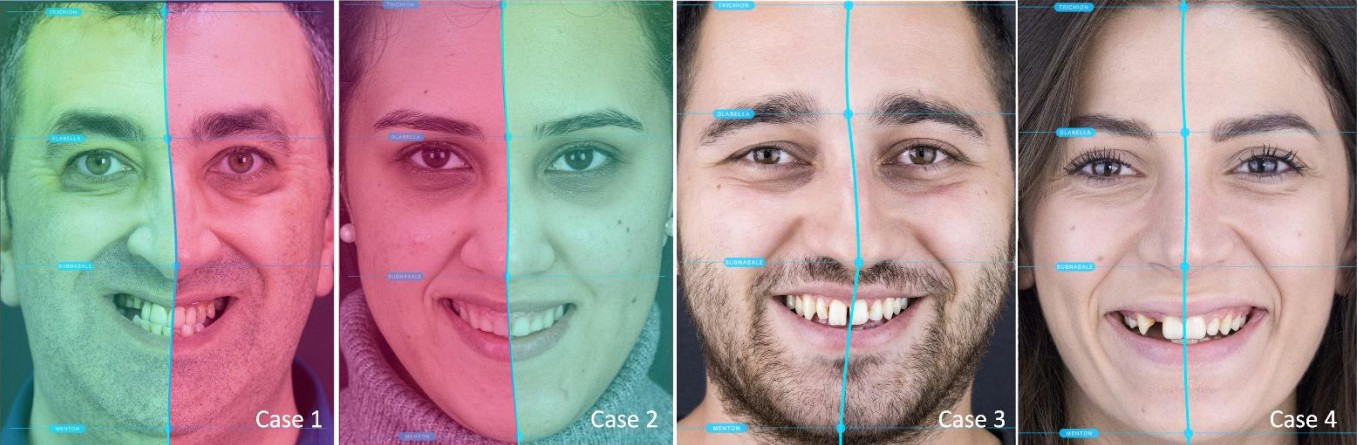

**Figure 1.** FF direction to the right side of the face (**Case 1**), FF direction to the left side of the face (**Case 2**), With the nose and chin pointing in different directions of the face (**Case 3**), A symmetrical face (**Case 4**).

### 2.2. Image Manipulation

Using the Smile Designer App program, two different smile designs were created for each case: one in the AI mode and one in the manual mode. The program uses the Microsoft Face API, a powerful AI-based tool that provides facial recognition capabilities. In a Node.js environment, the necessary browser-specific components, such as HTMLImageElement, HTMLCanvasElement, and ImageData, can be polyfilled. This can be achieved by installing the node-canvas package or, alternatively, by constructing tensors from image data and passing them as inputs to the API. The Microsoft Face API combines high-accuracy face recognition with advanced capabilities such as emotion detection, age and gender estimation, facial attribute analysis, real-time face detection and tracking, support for custom recognition models, and seamless cloud integration. These features open up a world of possibilities for diverse applications, ranging from personalized experiences and targeted marketing to augmented reality and virtual try-on experiences. The Microsoft Face API identifies 68 different facial landmarks on a person's face. These landmarks are crucial for the program's ability to locate and analyze various facial features, such as the position of the teeth. Once facial recognition is completed, the program calculates the distance between these 68 landmarks. This information is then used to position the teeth accurately. Based on the relationships between the facial landmarks, the program can determine the patient's facial type (e.g., square, triangle, round). This information is crucial in determining the best possible treatment and approach for the patient. With the information gathered from the facial landmarks, the program can automatically determine the appropriate tooth sizes for the patient, allowing for a more accurate and personalized treatment plan. A single physician with experience in using smile design programs created all of the designs. For each case, the reference points (trichion, glabella, subnasal, menton, pupillary, alare, and chellion regions) were automatically determined through facial analysis in the AI mode. DSDs were then created based on the different tooth shapes and sizes determined by considering the reference points.

In the manual mode, the physician subjectively selected and marked all reference points. Next, the teeth were selected in the AI mode and repositioned according to the reference points determined by the physician, without altering their color, shape, size, or form. The smile design was finally created (Figure 2).

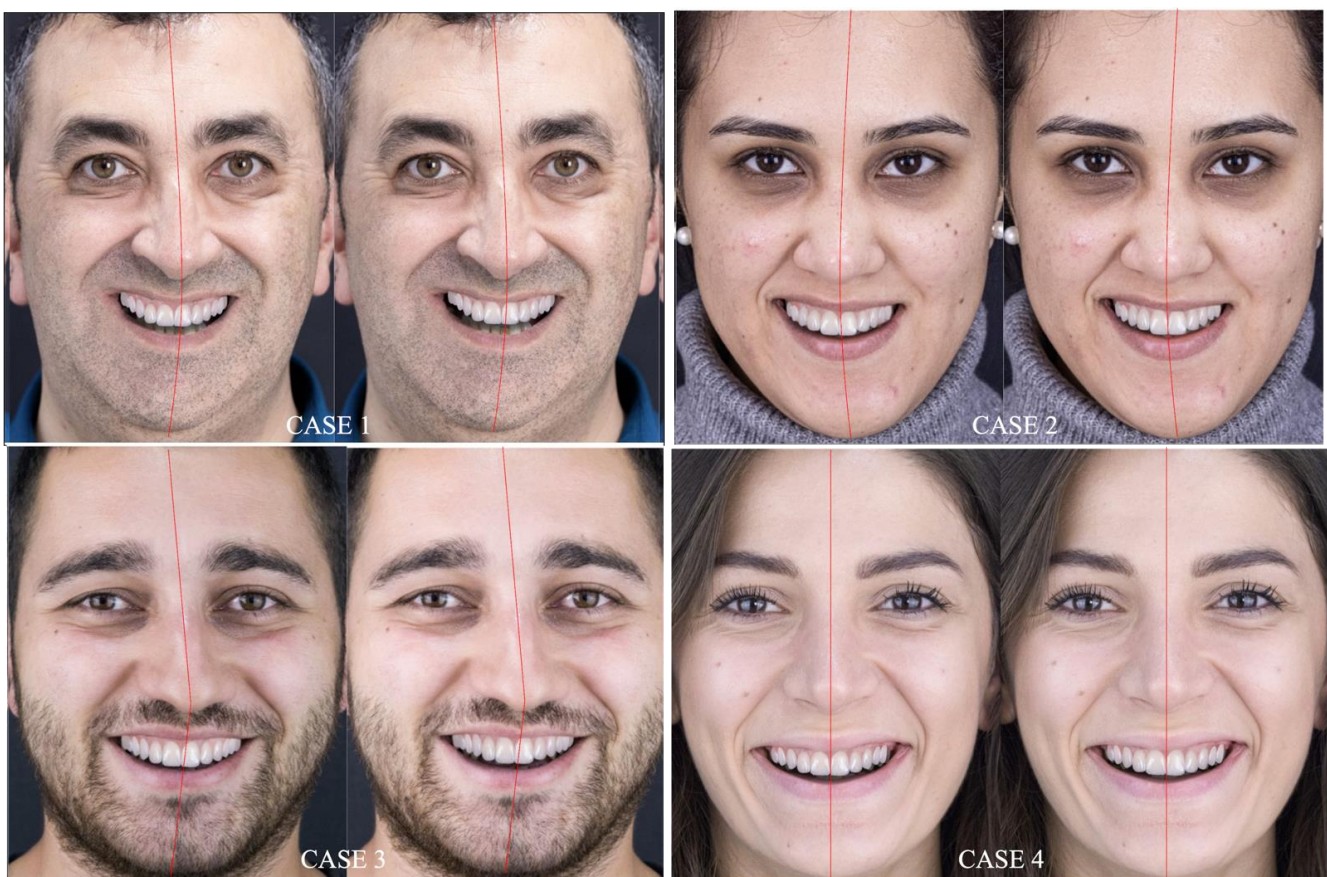

**Figure 2.** Artificial Intelligence mode (**left**), Manual mode (**right**) for all cases (Case photographs were used in the survey without red lines showing FF).

## 2.3. Preparation of Survey Questions and Administration of the Survey

An online survey (1999–2021, SurveyMonkey, California, CA, USA) was conducted to evaluate whether the manual or AI-based selection of landmarks on symmetrical and asymmetrical faces in smile design programs affects the esthetics of the resulting smile design. The survey link was shared online on social platforms and individuals under the age of 18 were not included in the study. After obtaining consent for voluntary participation in the survey, participants were asked to provide their age, sex, and occupation information. Participants were divided into three subgroups based on their occupation: dentist, dentistry student, and other professionals (laypeople). They were then asked about their professional knowledge and, for each professional group, about their experience and expertise in smile design. If participants had such knowledge and experience, they were asked whether they actively used a smile design program.

For each of the four cases, participants were asked to choose which of two smile designs, one created using AI and one created manually, they found more attractive. Only one option could be chosen in the online survey. Images were included with the options. After the surveys were completed, the data were transferred to a spreadsheet table (Excel 2021; Microsoft Corp., Washington, DC, USA) and analyzed statistically.

## 2.4. Statistical Analysis

The sample size was calculated using the "G*Power" Software (G*Power 3.1.9.213, Heinrich Heine Universitat Dusseldorf Institute Experimentelle Psychologie, Dusseldorf, Germany). The software output analysis reported 589 as a total sample size, with an under effect size of 0.2, an error probability alpha of 0.05, a power of 99%, and a confidence interval of 95%. After collecting the information from the participants via SurveyMonkey,

the data was entered into an excel spreadsheet (Excel 2021; Microsoft Corp). The IBM SPSS Statistics (IBM SPSS Statistics v22; IBM Corp.,Chicago, IL, USA) software package was used for the statistical analysis of the study. The chi-square test "Pearson's and Fisher's exact test (P)" was used to evaluate the degree of significance between the categorical variables.

## 3. Results

A total of 807 people confirmed that they had participated in the study voluntarily, and 628 participants who answered all questions were included in the study. The gender distribution of participants based on their choice between AI-generated and manually-created smile designs is shown in Table 1.

**Table 1.** Descriptive statistics for Manual and AI mode selections by gender.

| Case | Gender | Manual (%) | AI (%) | $p$ |
|------|--------|-----------|--------|-----|
| 1 | Female | 71.9 | 28.1 | 0.359 |
| | Male | 75.2 | 24.8 | |
| 2 | Female | 67.2 | 32.8 | 0.464 |
| | Male | 70 | 30 | |
| 3 | Female | 44.6 | 55.4 | 0.007 * |
| | Male | 55.7 | 44.3 | |
| 4 | Female | 44.6 | 55.4 | 0.611 |
| | Male | 46.7 | 53.3 | |

The chi-square test (* $p < 0.05$).

According to the data obtained from the sociodemographic survey questions, 330 dentists participated in the study. It was found that the majority of dentists had 0–4 years of experience (29.87%), while dentists with 25 or more years of experience made up only 4.87% of this group.

The majority of the dentists who participated in the survey did not have a specialty in any field (34.42%). The percentage of participants according to their area of expertise was as follows: prosthodontists (26.95%), orthodontists (10.06%), oral and maxillofacial surgeons (7.79%), endodontists (7.47%), restorative dentistry specialists (4.22%), periodontologists (3.90%), pedodontists (2.92%), and oral and maxillofacial radiologists (2.27%).

It was observed that 47.95% of laypeople and 59.06% of dentistry students who participated in the study stated that they had knowledge about esthetic smile design. The data were collected from a total of 149 students, 103 of whom were female (70.55%) and 46 of whom were male (29.45%). Of the 149 students, 113 had received preclinical education (18.12% in the 1st grade, 30.87% in the 2nd grade, and 26.85% in the 3rd grade), and 36 had practiced on patients (18.79% in the 4th grade and 5.37% in the 5th grade).

For Cases 1, 2, and 3, both dentists who used and did not use a smile design program preferred the manually-created smile design over the AI-generated smile design (preference percentages for Cases 1, 2, and 3 were 82.35%, 73.53%, and 64.71%, respectively, for those who had used a smile design program, and 79.56%, 72.63%, and 53.65%, respectively, for those who had not). For Case 4, dentists who used a smile design program preferred the manually-created smile design (55.88%), while those who did not use a smile design program preferred the AI-generated smile design (55.84%). There was no significant difference between the two groups of dentists for all of the cases ($p > 0.05$).

The chi-square test was used to evaluate the difference in esthetic preference for AI-generated or manually-created smile designs between different occupational groups. The results showed a significant difference in esthetic perception between the groups for Cases 1, 2, and 3 ($p = 0.002$, $p = 0.025$, and $p = 0.003$, respectively), but not for Case 4 ($p = 0.284$). A post-hoc test was conducted on the groups where a significant difference was found, and it was determined that there was a significant difference between dentists and dental

students ($p = 0.001$) and between dentists and laypeople ($p = 0.001$) for Case 1, only between dentists and dental students ($p = 0.003$) for Case 2, and only between dentists and laypeople ($p = 0.001$) for Case 3. In general, dentists preferred the manually created smile design in Cases 1, 2, and 3 (79.87%, 72.72%, and 54.87%, respectively) (Figure 3).

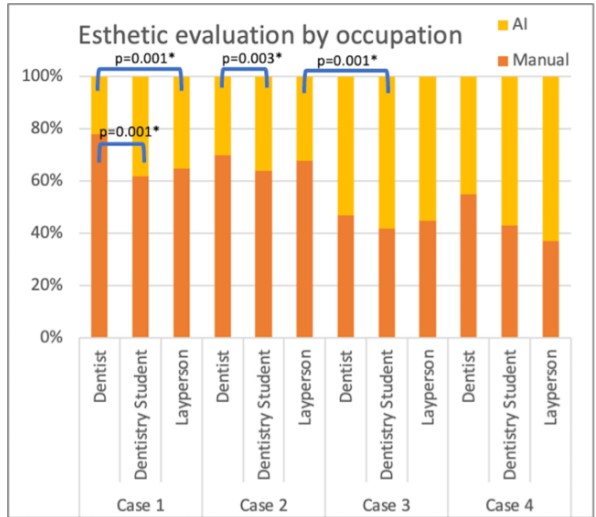

**Figure 3.** Comparison of participants by occupation, post hoc analysis after a Chi-square test, significant differences ($p < 0.05$) are indicated by asterisks.

When the preference rates for manually and AI-generated smile designs were evaluated according to the area of specialty or status of the dentists (expert or not), it was found that the difference between orthodontists and other specialties was statistically significant only for Case 3 ($p = 0.025$). Orthodontists generally preferred the AI-generated smile design (70.96%), while dentists in other specialties preferred the manually created design (51.88%) (Figure 4).

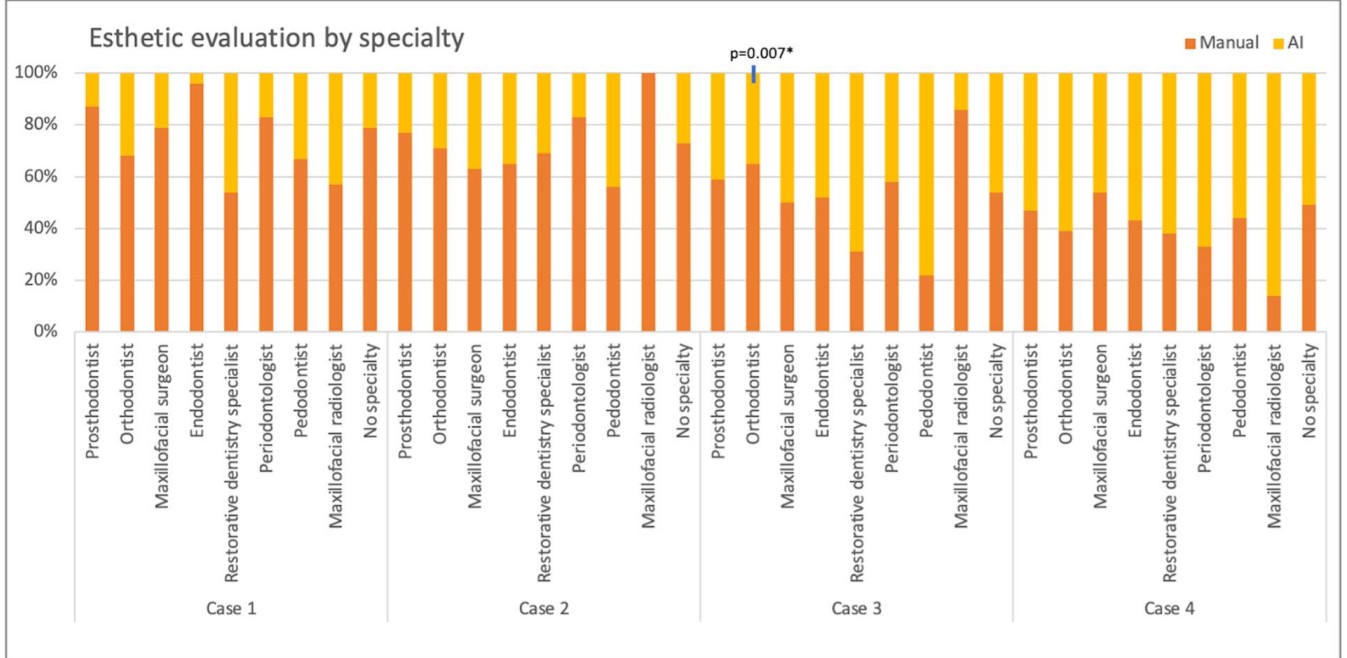

**Figure 4.** Comparison of participants by specialty, post hoc analysis after a Chi-square test, significant differences ($p < 0.05$) are indicated by asterisks.

When choosing between manually-created and AI-generated smile designs, the age of the individual (divided into ten groups ranging from 18–24 to 65 and above) (Figure 5), their education level (high school/middle school, associate degree, and undergraduate), and the clinical experience of the dentists (0–9 years, 10–19 years, and 20 years or more) were not found to be influential factors ($p > 0.05$). The chi-square test also showed that gender was not an influential factor in Cases 1, 2, and 4 ($p = 0.359$, $p = 0.464$, $p = 0.611$, respectively). However, for Case 3, a significant difference was observed between males and females, with females preferring AI-generated smile designs over manually-created ones ($p = 0.007$) (Figure 6).

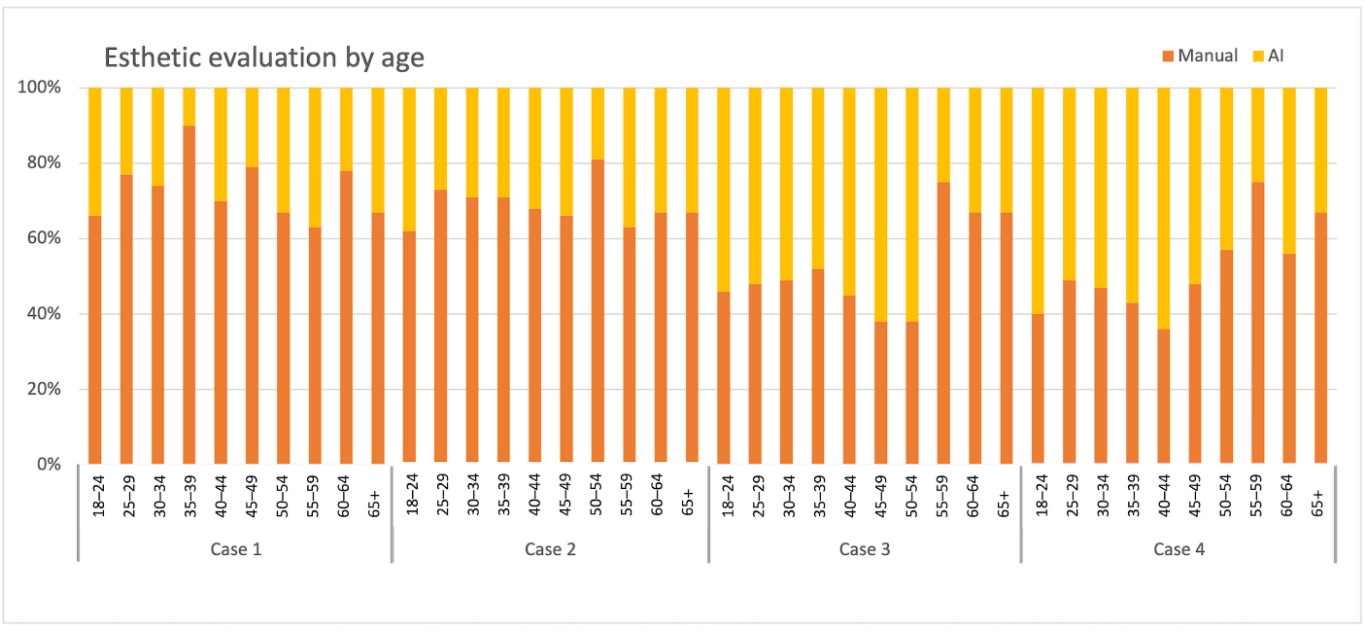

**Figure 5.** Comparison of participants by age.

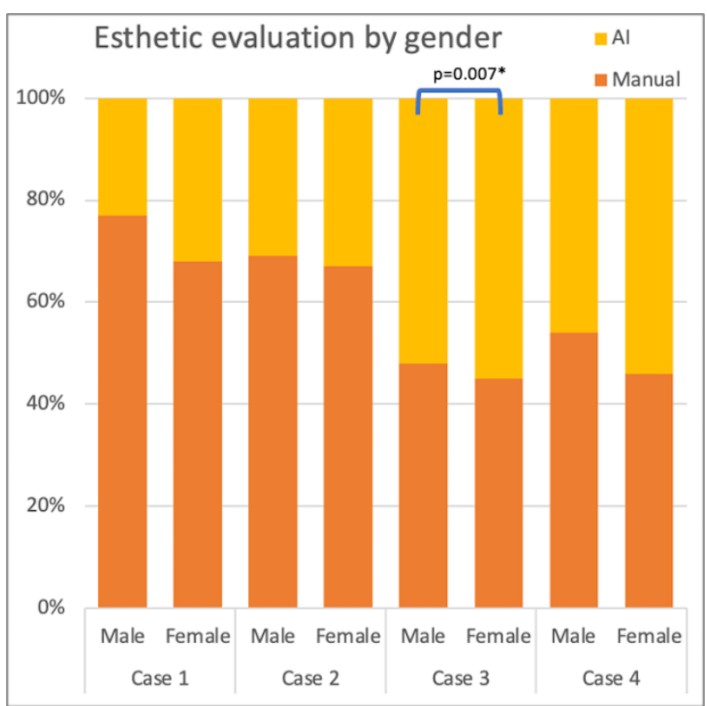

**Figure 6.** Comparison of participants by gender, post hoc analysis after a Chi-square test, significant differences ($p < 0.05$) are indicated by asterisks.

## 4. Discussion

Based on the results of this study, the null hypothesis, "the method of selecting landmarks manually or using AI in smile design programs does not affect the esthetic perception of the resulting smile design," was rejected. The study found significant differences in the esthetic perception of smile designs created manually or by AI.

As digital technology improves and people become more concerned with esthetics, DSD programs have become increasingly popular in dentistry [3,4,7,8]. Accurate diagnosis and treatment planning are crucial in esthetic dentistry, and DSD programs allow for virtual esthetic analysis and treatment planning using edited photographs or digital patient models [11].

Using 3D digital design can increase patient satisfaction and treatment success [4]. DSD programs provide; however, the ideal features that DSD programs should have are still being researched. A review of commonly used DSD programs found that if one or more esthetic parameters are neglected, the ideal treatment plan and results cannot be achieved. Therefore, it is believed that the selection of landmarks affects treatment planning and the esthetics of the restoration created by the program [5].

In this study, the landmarks were selected manually by the clinician and automatically by AI. The automatic selection of landmarks by AI saves time, but the fact that dentists generally found the cases with manually selected landmarks to be more esthetically pleasing raises questions about the effectiveness of AI.

For the most accurate determination of symmetry and asymmetry in facial analysis, as many reference points as possible should be used [24]. In the AI mode of the smile design program used in this study, 68 facial points were determined using Microsoft face API.

It was observed in Case 3 that orthodontists preferred the AI-generated smile design at a statistically significant rate compared to other specialties. This may be due to the fact that orthodontists use FM more frequently during treatment, leading to their heightened perception of smiles [12]. The reason orthodontists had different esthetic perceptions in our study may be attributed to this. In a study comparing landmarks marked by orthodontists to those drawn by AI, a high degree of similarity was found [17]. This suggests that orthodontists may have preferred the AI designs in our study due to their familiarity with AI values or experience with treatments based on these familiar landmarks [18].

In contrast, when examining Case 4 (which featured a symmetrical facial feature), there was no significant difference in preference for manually created or AI-generated designs among dentists and other occupations ($p = 0.001$). It can be concluded that in complex cases, dentists' esthetic perception is different from that of people in the non-professional group [13].

The results of the survey suggest that there is a significant difference in the esthetic perception between dentists and laypeople, similar to findings from another study [1]. This indicates that there may be a discrepancy between the esthetic perceptions of patients and dentists during treatment.

For years, clinicians, researchers, and programmers have favored mathematical approaches for creating symmetrical smiles. However, the FF concept allows for more organic and natural smile designs on both symmetrical and asymmetrical faces. While traditional smile design methods often aim for symmetry by using straight lines, such as the dental midline, as a reference, the FF concept takes into account human perception and may be more suitable for asymmetrical faces [19]. In DSD applications, mathematical algorithms are used to create a new design based on determined landmarks. While AI and DSD programs may produce good results in patients with good facial symmetry, manual methods may be preferred in cases where facial symmetry is impaired. Further research on the use and effectiveness of AI in creating DSDs could help clinicians understand the relationship between landmarks and achieve the ideal smile design, as well as understand human esthetic perception. More studies are needed to determine the best approaches for creating natural and harmonious smiles, and to decide whether facial structures should be altered with a new smile design.

In terms of methodology, this study is the first to evaluate how individuals perceive smile designs created using both AI and manual design in terms of esthetics. While the results of the study provide useful information, there are some limitations to consider. The research was conducted online and the participants were classified into categories based on their occupations. However, due to the random nature of the online survey, it was not possible to include the same number of participants in each group. This means that a more homogeneous and specific distinction of categories may provide more information about an individual's esthetic perception. Another limitation of the study is that the compatibility of the current "smile design algorithm" used in the smile design program with other 2D and 3D smile design programs is not known.

Future studies should compare the AI mode of different smile design programs to further understand the relationship between AI and manual design in terms of esthetics. Furthermore, future studies should consider comparing the AI modes of different smile design programs to gain a deeper understanding of the relationship between AI and manual design in terms of esthetics. Additionally, we acknowledge that esthetics is subjective and significantly influenced by environmental factors, such as race and culture. Therefore, it is recommended that future research explores the impact of conducting similar studies in various demographic regions to determine how these factors affect esthetic evaluations.

## 5. Conclusions

Based on the limitations of this observational study, it was found that age, education level, and clinical experience did not significantly affect individuals' esthetic preferences for smile designs created either manually or by AI. However, when evaluating the esthetic perception of smile designs created manually or by AI, significant differences were observed between genders, professions, and specialties in both symmetrical and asymmetrical situations. In cases where AI-generated smile designs were found to be preferred or comparable to manually-created designs, their use in symmetrical faces could be considered acceptable to both dentists and laypeople, potentially saving time for clinicians.

**Author Contributions:** Conceptualization, G.C. and G.S.Ö.; Data curation, G.M.; Formal analysis, G.C. and G.M.; Investigation, G.C., G.S.Ö. and F.E.; Methodology, G.C. and G.S.Ö.; Software, S.Ş.; Validation, G.C. and G.M.; Visualization, G.C. and G.S.Ö.; Writing—original draft, G.C., G.S.Ö. and F.E.; Writing—review and editing, G.C., F.E. and S.Ş. All authors have read and agreed to the published version of the manuscript.

**Funding:** This research received no external funding.

**Institutional Review Board Statement:** The protocol of this study was approved by the Ethics Committee of Istanbul Medipol University (approval number: 2020/561).

**Informed Consent Statement:** All participants and cases provided their consent to participate voluntarily in the online survey. In addition, signed consent forms were obtained from the cases whose images were used in the digital smile designs included in the study, in order to allow for the use of their images for open-access publication.

**Data Availability Statement:** The data presented in this study are available on request from the corresponding author.

**Acknowledgments:** We thank Elif Beycan Şen for helping to create the Digital Smile Designs used in this study.

**Conflicts of Interest:** The authors declare no conflict of interest.

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
