# Peer review of "Evaluating the Facial Esthetic Outcomes of Digital Smile Designs Generated by Artificial Intelligence and Dental Professionals"

_applsci, doi:10.3390/app13159001_

Round 1

Reviewer 1 Report

The topic of the paper is interesting, and the writing of the manuscript is good. However, the reviewer has some comments as follows.

(1)  The contribution of the authors seems to be little, but only an online survey.

(2)  The dataset collected from the survey is too small for discovering a meaningful conclusion.

(3)  The analyses to the experiment results seem to be solid, but not so easy for readers to find valuable conclusion.

good

Author Response

Dear Reviewer,

We sincerely appreciate the reviewer's comments and suggestions, which have greatly helped improve the manuscript's quality. We have corrected and modified the manuscript according to the comments given by the reviewer. The authors have approved the revisions. The changes are visible in the original paper highlighted in yellow. The responses are given below.

Point 1: The contribution of the authors seems to be little, but only an online survey.

Response 1: We appreciate the reviewer's input. We would like to reiterate that the authors' contributions are accurate as stated in the manuscript. Throughout the study, each author fulfilled their respective roles in multiple stages, including research planning, design, case selection, utilization of the smile design program for creating designs, formulation of survey questions, distribution of the online survey through social platforms, data collection, statistical analysis, result interpretation, and the composition of the original main text.

Point 2: The dataset collected from the survey is too small for discovering a meaningful conclusion.

Response 2: We sincerely appreciate the valuable contribution made by the reviewer. In order to determine an appropriate sample size, we used the 'GPower' Software (GPower 3.1.9.213, Heinrich Heine Universitat Dusseldorf Institute Experimentelle Psychologie, Dusseldorf, Germany). The analysis using this software indicated a total sample size of 589, with an effect size of 0.2, an alpha error probability of 0.05, a power of 99%, and a 95% confidence interval. The questions posed to participants, including their age, gender, occupation, specialty, class (for dentistry students), and familiarity with the smile design program and its usage, were crucial for the initial planning of our study.

The objective of our study was to evaluate the aesthetic perception of facially driven smile designs, comparing those created manually with those generated using AI. Our initial hypothesis posited that the method of selecting landmarks (manual or AI-based) in smile design programs would not significantly impact the aesthetic perception of the resulting smile design, regardless of facial symmetry. Through our survey, we gathered data on participants' age, gender, profession, specialty (for dentists), class (for dentistry students), and familiarity with digital smile design as a comparative criterion. Hence, the dataset we obtained aligns with the information presented in the manuscript.

Point 3: The analyses to the experiment results seem to be solid, but not so easy for readers to find valuable conclusion.

Response 3: We appreciate the reviewer's insightful comments and the opportunity to address their concerns. We understand their observation that it may be challenging for readers to extract valuable conclusions from our analyses. In light of this, we would like to further elucidate the importance and significance of our study as the initial research in the field of AI-designed smile designs.

Our study represents a pioneering effort to explore the impact of artificial intelligence on smile design in dentistry. As one of the first investigations in this emerging field, our research holds intrinsic value in setting the stage for future advancements and innovations. By venturing into uncharted territory, we have laid a solid foundation upon which further research and development can be built.

The scarcity of existing literature specifically focused on AI-designed smile designs underscores the significance of our study. By conducting this initial research, we have successfully addressed a crucial gap in the current knowledge landscape. Our findings contribute to the growing body of literature and serve as a starting point for further exploration, refinement, and optimization of AI algorithms in the context of dental esthetics.

Beyond the academic realm, our study holds practical implications for dental practice. The integration of AI in smile design programs has the potential to revolutionize the field by enhancing efficiency, accuracy, and patient satisfaction. Patients' ability to visualize the esthetic outcome of their treatment through digital smile designs amplifies the prevalence of these applications in clinical settings. Moreover, the time-saving advantages offered by AI-driven smile design programs can greatly benefit both clinicians and patients.

In conclusion, we are grateful for the reviewer's valuable consideration and are committed to addressing the concern regarding the accessibility of valuable conclusions. Through our pioneering research, we have paved the way for future investigations, addressed a research gap, and opened doors to revolutionary advancements in the field of AI-designed smile designs. We will strive to present our findings in a clear and concise manner, ensuring that readers can easily grasp the significant contributions of our study.

We hope to have addressed the reviewer’s concerns satisfactorily.

Sincerely

Reviewer 2 Report

1. It seems that the method of classifying into four cases  shown in Fig. 1 is not presented in detail. In addition, the classification method seems to be affected by the direction of the photographing, but it is hoped that an explanation for this is mentioned.

2. Since AI is greatly affected by the data it learns, it is recommended to provide an explanation of what data the AI used was trained with.

3. There seems no difference between the pictures shown in Fig. 2 and they all look the same.

4. Esthetics is subjective and is greatly influenced by environmental factors such as race and culture, so it is requested that the data used for the preference survey include information about these factors or the degree to which these factors affect the evaluation of esthetics.

Author Response

Dear Reviewer,

We sincerely appreciate the reviewer's comments and suggestions, which have greatly helped improve the manuscript's quality. We have corrected and modified the manuscript according to the comments given by the reviewer. The authors have approved the revisions. The changes are highlighted in yellow in the original paper. The responses are given below.

Point 1: It seems that the method of classifying into four cases shown in Fig. 1 is not presented in detail. In addition, the classification method seems to be affected by the direction of the photographing, but it is hoped that an explanation for this is mentioned.

Response 1: We appreciate the reviewer's valuable contribution. In response to the concern, we have provided a more detailed explanation of the classification method used for the cases in the Materials and Methods section. We believe that these additional details provide a comprehensive explanation of the classification method employed, addressing the reviewer's concerns and enhancing the clarity of our study.

Revised text 1: The classification of cases was based on the relationships between reference points on the forehead, nose, and chin tip, which are significant in smile designs. Specifically, the trichion, glabella, subnazale, and menton points were utilized as references to determine the facial flow. By examining the axis formed by connecting these points, we identified four distinct cases: Case 1 represented a facial flow towards the right side, Case 2 represented a facial flow towards the left side, Case 3 represented a scenario where the nose and chin pointed in different directions, and Case 4 represented a symmetrical face (Figure 1). To aid visual clarity, we utilized a color scheme in Figure 1, where the side with the facial flow direction was depicted in green, while the opposite side was shown in red. However, it is important to note that in neutral cases where the nose and chin tips were directed to different sides while maintaining facial symmetry, no green or red sides were indicated [19].

 Point 2: Since AI is greatly affected by the data it learns, it is recommended to provide an explanation of what data the AI used was trained with.

Response 2: We would like to thank the reviewer for this important criticism. We have given an explanation about the artificial intelligence algorithm used by the program in the Materials and Methods section. We appreciate the reviewer's request for further information on the capabilities of the Microsoft Face API. In response to their query, we would like to provide a comprehensive overview of the API's features and benefits.

Face recognition accuracy: The Microsoft Face API excels in accurately detecting and recognizing faces in images and videos. It demonstrates robustness in handling challenging conditions, such as varying lighting, different angles, diverse facial expressions, and partial occlusions. Its advanced algorithms ensure reliable face recognition performance.

Emotion detection: The API goes beyond facial recognition by analyzing facial expressions and detecting a wide range of emotions, including happiness, sadness, anger, surprise, and more. This capability opens doors to applications such as sentiment analysis, user engagement measurement, or personalized experiences based on emotional responses.

Age and gender estimation: Leveraging its advanced algorithms, the Microsoft Face API can estimate the age and gender of individuals based on their facial features. This feature allows for demographic analysis, targeted marketing campaigns, or age-specific user experiences in various industries.

Facial attribute analysis: With the ability to extract detailed facial attributes, including facial hair, glasses, makeup, and more, the API offers valuable insights for personalized recommendations, virtual try-on experiences, or beauty and cosmetic applications. This enhances user engagement and fosters interactive and tailored experiences.

Real-time face detection and tracking: The Microsoft Face API provides real-time face detection and tracking capabilities, making it suitable for applications requiring immediate responses. Whether it's augmented reality, virtual makeup, or gesture recognition, the API enables seamless and interactive experiences in real-time video streams.

Custom recognition models: The flexibility of the Microsoft Face API allows developers to train their own custom face recognition models. By tailoring the models to specific requirements or datasets, developers can create unique and specialized face recognition solutions that address their specific needs, enhancing accuracy and performance.

Scalability and cloud integration: As a cloud-based service, the Microsoft Face API offers scalability and accessibility from anywhere. It seamlessly integrates with other Microsoft Azure services, providing a comprehensive ecosystem for building powerful applications. The API's compatibility with various programming languages and platforms ensures easy integration and wide usability.

In conclusion, the Microsoft Face API combines high accuracy face recognition with advanced capabilities such as emotion detection, age and gender estimation, facial attribute analysis, real-time face detection and tracking, support for custom recognition models, and seamless cloud integration. These features open up a world of possibilities for diverse applications, ranging from personalized experiences and targeted marketing to augmented reality and virtual try-on experiences.

Revised text 2: Using the Smile Designer App program, two different smile designs were created for each case: one in the AI mode and one in the manual mode. The program uses the Microsoft Face API, a powerful AI-based tool that provides facial recognition capabilities. In a Node.js environment, the necessary browser-specific components, such as HTMLImageElement, HTMLCanvasElement, and ImageData, can be polyfilled. This can be achieved by installing the node-canvas package or, alternatively, by constructing tensors from image data and passing them as inputs to the API. The Microsoft Face API combines high-accuracy face recognition with advanced capabilities such as emotion detection, age and gender estimation, facial attribute analysis, real-time face detection and tracking, support for custom recognition models, and seamless cloud integration. These features open up a world of possibilities for diverse applications, ranging from personalized experiences and targeted marketing to augmented reality and virtual try-on experiences. The Microsoft Face API identifies 68 different facial landmarks on a person's face. These landmarks are crucial for the program's ability to locate and analyze various facial features, such as the position of the teeth. Once facial recognition is completed, the program calculates the distance between these 68 landmarks. This information is then used to position the teeth accurately. Based on the relationships between the facial landmarks, the program can determine the patient's facial type (e.g., square, triangle, round). This information is crucial in determining the best possible treatment and approach for the patient. With the information gathered from the facial landmarks, the program can automatically determine the appropriate tooth sizes for the patient, allowing for a more accurate and personalized treatment plan.

Point 3: There seems no difference between the pictures shown in Fig. 2 and they all look the same.

Response 3: To ensure the main text remains concise, we presented eight photographs of the four cases within a single figure. However, we recognize the importance of providing clear visual representations for each case in the survey questions. Therefore, we have included an enlarged photograph below to illustrate the differences between smile designs created by artificial intelligence and manual methods, presented side by side on a single page.

Importantly, the results of our study indicate that there is minimal difference between the smile designs created for symmetrical faces. In agreement with the reviewer's observation, we found no significant distinction between the photographs in the case of symmetrical faces. This finding is consistent with our study's results.

By including the enlarged photograph and acknowledging the lack of substantial differences in smile designs for symmetrical faces, we aim to provide a clearer understanding of the visual comparisons while effectively addressing the reviewer's comments.

 Point 4: Esthetics is subjective and is greatly influenced by environmental factors such as race and culture, so it is requested that the data used for the preference survey include information about these factors or the degree to which these factors affect the evaluation of esthetics.

Response 4: We appreciate the reviewer's comment and agree with their suggestion. In response to this valuable feedback, we have made the necessary additions to the discussion section of our manuscript. These additions now address the limitation and provide suggestions for future research.

Revised text 4: Furthermore, future studies should consider comparing the AI modes of different smile design programs to gain a deeper understanding of the relationship between AI and manual design in terms of esthetics. Additionally, we acknowledge that esthetics is subjective and significantly influenced by environmental factors, such as race and culture. Therefore, it is recommended that future research explores the impact of conducting similar studies in various demographic regions to determine how these factors affect esthetic evaluations.

By incorporating these revisions, we have acknowledged the reviewer's suggestion, highlighted the importance of further research, and acknowledged the influence of race and culture on esthetic evaluations.

We hope to have addressed the reviewer’s concerns satisfactorily.

Sincerely

Reviewer 3 Report

This article evaluates the preference rates for smile designs created by different people and suggests that the use of AI-generated smile designs for symmetric faces is acceptable to both dentists and laypeople and can offer time-saving benefits for clinicians. 

The issues about the content of the manuscript include:

1. The study does not provide information on the racial or ethnic diversity of the participants, which limits the ability to draw conclusions about the preferences of different demographic groups.

2.  It should be directly illustrated that "P" is the abbreviation of "Pearson's and Fisher's exact test". For example, "Pearson's and Fisher's exact test (P)"

3.  Showing the line of FF of Figure 2. to better instruct the difference with Figure 1.

4.  Make the words size of Figure 3 and Figure 6 smaller.

no

Author Response

We appreciate the reviewer for their valuable comments and contributions.

We hope to address the reviewer's comments in the response letter adequately.

Sincerely

Round 2

Reviewer 2 Report

The revised paper has been improved to closely match the referee's comments.

Author Response

We sincerely appreciate the reviewer's comments and suggestions, which have greatly helped improve the manuscript's quality. Sincerely
